# Cryo-EM structure supports a role of AQP7 as a junction protein

Peng Huang ®[1], Raminta Venskutonytė ®[1,2], Rashmi B. Prasad ®[3], Hamidreza Ardalani ®[4], Sofia W. de Maré[1], Xiao Fan[5], Ping Li[1], Peter Spégel ®[4], Nieng Yan ®[5], Pontus Gourdon ®[1], Isabella Artner ®[3] & Karin Lindkvist-Petersson ®[1,2] ✉

Aquaglyceroporin 7 (AQP7) facilitates glycerol flux across the plasma membrane with a critical physiological role linked to metabolism, obesity, and associated diseases. Here, we present the single-particle cryo-EM structure of AQP7 determined at 2.55 Å resolution adopting two adhering tetramers, stabilized by extracellularly exposed loops, in a configuration like that of the well-characterized interaction of AQP0 tetramers. The central pore, in-between the four monomers, displays well-defined densities restricted by two leucine filters. Gas chromatography mass spectrometry (GC/MS) results show that the AQP7 sample contains glycerol 3-phosphate (Gro3P), which is compatible with the identified features in the central pore. AQP7 is shown to be highly expressed in human pancreatic α- and β- cells suggesting that the identified AQP7 octamer assembly, in addition to its function as glycerol channel, may serve as junction proteins within the endocrine pancreas.

Aquaporins (AQPs) are channel proteins that are responsible for water and glycerol regulation in the human body. Besides solute transport, AQPs have been shown to be involved in cellular signaling, cell migration, and cell–cell adhesion[1]. In rodents, AQP7 is the main glycerol channel in the endocrine pancreas, mediating rapid flux of glycerol in β-cells and ultimately insulin secretion[2,3]. Recently it was shown that AQP7 contributes to cell–cell interactions within cells of the islet of Langerhans, and impaired AQP7 expression was connected to pancreatic β-cell dysfunction[4], however reports on the role of AQP7 in human pancreas are scarce.

Here we have analyzed *AQP7* gene expression in human islets from 188 donors and observed mRNA expression of *AQP7* in α-, β-, and δ-cells. Single nucleotide polymorphisms (SNP) at the *AQP7* locus were strongly associated with type II diabetes (T2D). In addition, we report a 2.55 Å single-particle cryo-EM structure of human AQP7, revealing two adhering tetramers. The assembly interacts via extracellularly exposed loops, and thus provides a molecular basis for adhesion of AQP7-exposing cells. In-between the four monomers of each tetramer a

relatively wide central pore is identified. Well-defined densities are observed in the central pore that are structurally compatible with one of the components present in the protein sample, glycerol 3-phosphate (Gro3P). Pancreatic β-cells maintain a consistent orientation with respect to the islet capillaries[5], and interestingly AQP7 is known to also be highly expressed in capillaries[6]. Thus, here we propose an unreported role for human AQP7 within the islet of Langerhans, where it functions as junction proteins to provide cell adhesion as well as supply of glycerol and other metabolites.

## Results

### AQP7 is expressed in human pancreatic islets and genetic variants of AQP7 are associated with blood glucose control and type II diabetes

Previous reports have shown that AQP7 is expressed in rodent pancreatic islets and that expression is reduced in animal models of T2D[3,7]. To evaluate if AQP7 is present in human islets, immunohistochemistry using AQP7 and islet hormone antibodies was performed on

---

[1]Department of Experimental Medical Science, Lund University, Lund, Sweden. [2]LINXS—Lund Institute of Advanced Neutron and X-ray Science, Lund, Sweden. [3]Lund University Diabetes Centre, Clinical Research Center, Malmo, Sweden. [4]Centre for Analysis and Synthesis, Department of Chemistry, Kemicentrum, Lund University, Lund, Sweden. [5]Department of Molecular Biology, Princeton University, Princeton, NJ, USA. ✉e-mail: karin.lindkvist@med.lu.se

pancreatic sections from normoglycemic donors. AQP7 protein was co-expressed with insulin, glucagon, and somatostatin (Fig. 1a). A similar expression pattern was also obtained from single cell transcriptomics of human islets, confirming that *AQP7* is expressed in human pancreatic α-, β-, and δ-cells (Fig. 1b). Next, we assessed if *AQP7* expression levels are altered in islets from T2D donors. RNAseq analysis from 188 donors showed that *AQP7* transcripts were decreased in islets from T2D donors (Fig. 1c). In addition, a negative correlation between *AQP7* expression in islets and BMI was detected (Fig. 1d) suggesting that islet *AQP7* expression is associated with islet function/ whole body metabolism. To evaluate if genetic variants in *AQP7* gene locus were associated with T2D and metabolic traits associated with blood glucose control (HbA1c, fasting and random blood glucose measurements), genome wide association data look ups were made. The rs2247654 variant in the *AQP7* gene showed strong signals of association with random glucose ($p = 1.52*10^{-12}$) and fasting glucose adjusted for BMI ($p = 5.08*10^{-12}$, AMPT2D portal: hugeamp.org, Supplementary Fig. 1a)[8–10]. The SNPs rs83921 (~0.4 Mb distance of *AQP7*) and rs855532 (~0.25 Mb distance from *AQP7*) were significantly associated with T2D ($p = 2.39*10^{-9}$)[11] and HbA1c ($p = 7.42*10^{-8}$), respectively (https://www.kp4cd.org/node/120; AMPT2D portal: hugeamp.org, Supplementary Fig. 1b). This indicates that variants in the *AQP7* gene are associated with metabolic traits controlling physiological blood glucose levels and T2D, supporting an essential role for AQP7 in islet cell function.

### The overall structure of human AQP7

Human AQP7 was expressed in *Pichia pastoris*, solubilized in GDN detergent, and purified using affinity and size-exclusion chromatography (Supplementary Fig. 2a). Interestingly, the chromatographic profiles suggest different oligomeric states are present in a concentration-dependent manner (Supplementary Fig. 2b). Cryo-EM micrographs and related two-dimensional (2D) class averages show that AQP7 protein particles can form dimers of tetramers (Supplementary Fig. 2c). To rule out the possibility that AQP7 dimers of tetramers are induced by grid chemistry, human AQP7 protein samples were prepared in another type of grid as well, and the formation of dimers of tetramers was reproduced (Supplementary Figs. 3, 4a). AQP7 particles were picked and extracted from collected micrographs, followed by in silico 2D and 3D classification in cryoSPARC[12]. The particles in octamer class (~47% of all particles) were selected and reconstructed to 2.55 Å with D4 symmetry and 3.0 Å without symmetry application (Supplementary Figs. 3 and 5, Supplementary Table 1). Consequently, high-resolution features such as side-chain densities can be observed in both maps unambiguously (Supplementary Fig. 6). Overall, the differences between the two models are minor, with a root-mean-square deviation (RMSD) of 0.20 Å. Unless indicated otherwise, the structural analysis discussed below is based on the 3.0 Å map and model to avoid potential bias originating from the applied symmetry. The final maps reveal an assembly with the dimension of ~10 nm in height and 6 nm in diameter, displaying a double-layer density corresponding to two back-to-back AQP7 tetramers, separated by an ~3.3 nm long gap between the extracellular sides of the two interacting AQP7 tetramers (Fig. 2a, b, Supplementary Fig. 7). Interestingly, the two tetramers have a ~45° twisted arrangement relative to the central axis, leading to the formation of a stable octamer (Fig. 2c).

We were able to model residues 25-277 in the 2.55 Å consensus map, which includes the six transmembrane helices (TM1-6), the two half helices (HX1 and HX2) dipping into the plasma membrane, and five loops (A-E) connecting the transmembrane segments (Fig. 3a, Supplementary Fig. 2d). In addition, two conserved features of the aquaporin family are well defined in the maps, the aromatic/arginine (ar/R) selective filter and the NPA-motifs, the latter being replaced by NAA and NPS in AQP7 (Fig. 3a). The intracellular termini could not be fully modeled possibly due to their inherent flexibility, but a well-defined

fraction of the N-terminus extends from the periphery of the tetramer at the cytoplasmic side and is in close proximity to the C-terminus of the adjacent monomer (Fig. 3b). Specifically, R35 from the N-terminus forms a hydrogen bond with the backbone of F275 on the C-terminus of the adjacent monomer, contributing to the stabilization of the tetrameric configuration (Fig. 3b).

### AQP7 forms a dimer of tetramers using loop C

As clearly observed from the map and model, the two AQP7 tetramers are associated by the protruding extracellular C loops (Figs. 2a and 4a). Compared to orthodox aquaporins (strict water channels without glycerol conductance capacity), AQP7 harbors a ~10 residues extended extracellular loop C between TM3 and TM4 that features two short helices (Fig. 4a, Supplementary Figs. 2d, 8). In the single-particle cryo-EM structure determined here, loop C is protruding from the compact monomer and packs against the opposing monomer from the other tetramer, thus responsible for the formation of the octameric arrangement (Fig. 4a). Interactions between P151 and V152 (<4 Å apart) are detected between the two tetramers (Fig. 4a). In addition, Q145 in loop C forms a hydrogen bond with its opposing Q145 from the other tetramer (Fig. 4a). Notably, although all human aquaglyceroporins (AQP3, AQP7, AQP9 and AQP10) harbor an extended C loop, which contrasts with the orthodox aquaporins that have a much shorter loop C (Supplementary Fig. 8), Q145, P151 and V152 are not strictly conserved within the human aquaglyceroporin family (Fig. 4b). To investigate the importance of these residues for the octameric arrangement, the propensity for AQP3 (that has an asparagine at position 152 instead of a valine, Fig. 4b) to form dimers of tetramers was analyzed. Accordingly, two-dimensional (2D) class averages showed that AQP3 protein particles cannot form dimers of tetramers in same conditions as for AQP7 (Supplementary Fig. 4b), suggesting that V152 contributes to the dimer formation for AQP7. Nonetheless, previous reports have shown that other aquaporins can form dimer of tetramers utilizing different residues than AQP7, and act as junction/ adhesion proteins with a proposed similar tetramer-tetramer setup as observed here for AQP7 (Supplementary Fig. 9)[13,14].

### Glycerol is present in the channels

In accordance with the X-ray structure of AQP7[15], every monomer in the cryo-EM structure forms a glycerol channel, resulting in eight glycerol channels for the whole octameric structure (Fig. 5a). Densities corresponding to bound glycerol molecules are present in all individual channels in the symmetric and asymmetric cryo-EM reconstructions but to a varying degree (Fig. 5a). Still, cryo-EM density nearby the NPA-motif is present in all glycerol channels, suggesting a strong binding between glycerol and the asparagines, N94 and N226. This is in line with the X-ray structure where glycerols were suggested to interact with both asparagines[15]. HOLE analyses were performed for each individual chain (A–H) from 3.0 Å (C1 symmetry) cryo-EM model, the consensus monomer from 2.55 Å (D4 symmetry) cryo-EM model and the AQP7 crystal structure (Fig. 5a), and the relative glycerol channel radii were plotted (Fig. 5b, Supplementary Fig. 10). All the individual glycerol channels within the cryo-EM model display similar radii along the glycerol channels, and accordingly upon structural superposition the RMSD is below 0.2 Å between all the monomers (Supplementary Table 2). However, all the cryo-EM glycerol channels present a narrower ar/R filter and wider region at the position of NPA motif than that of the available crystal structure, suggesting dynamic properties of the glycerol channels (Fig. 5a, b, Supplementary Fig. 10).

### Ligands are present in the central pore

Permeation and selectivity through the water/glycerol channels of orthodox aquaporins and aquaglyceroporins are well-characterized, whereas the central pore of AQPs is poorly understood and it remains unclear whether it has a physiological role. Lined by non-polar residues

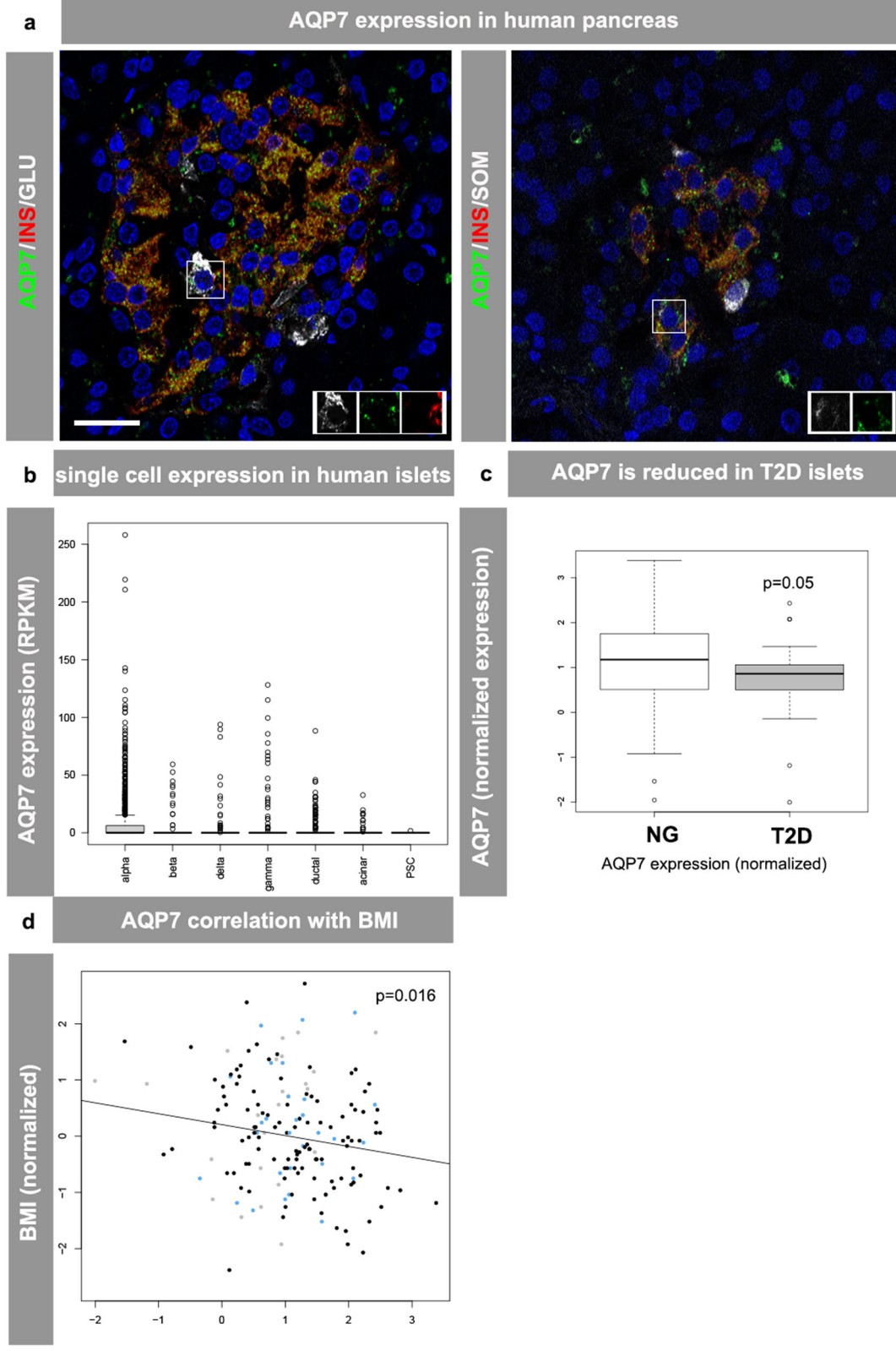

**Fig. 1 | AQP7 expression in human pancreatic islets.** Immunohistochemistry staining in human pancreas sections from normoglycemic donors for (**a**) left: AQP7 (green), insulin (red), and glucagon (white), right: AQP7 (green) and insulin (red), somatostatin (white). Nuclei are blue, scale bar is 20 μm, inserts illustrate co-expression of AQP7 with insulin and glucagon (left panel) and somatostatin (right panel). **b** Single cell RNAseq expression of AQP7 in human pancreatic cells from[37] (*n* = 10: NG = 6, T2D = 4). **c** AQP7 expression in islets from normoglycemic donors and donors diagnosed with T2D (NG = 155, T2D = 33 donors). All box plots include the median line, the box denotes the interquartile range (IQR), whiskers denote the rest of the data distribution and outliers are denoted by points >±1.5 × IQR. Differential expression between T2D and NG donors was calculated on TMM normalized data using linear models with lmFit, and *P*-values were calculated using the eBayes function in limma, implemented in edgeR. logFC = −0.33. **d** Negative correlation of islet AQP7 expression with body mass index (BMI), normoglycemic, glucose intolerant, and type 2 diabetic donors are shown in black, blue, and gray, respectively. *n* = 188 for (**c**, **d**). (statistics: Spearman correlation). *p*-values are adjusted for multiple testing using Benjamini–Hochberg correction (**c**, **d**). Source data underlying **b**–**d** are provided in the Source Data file.

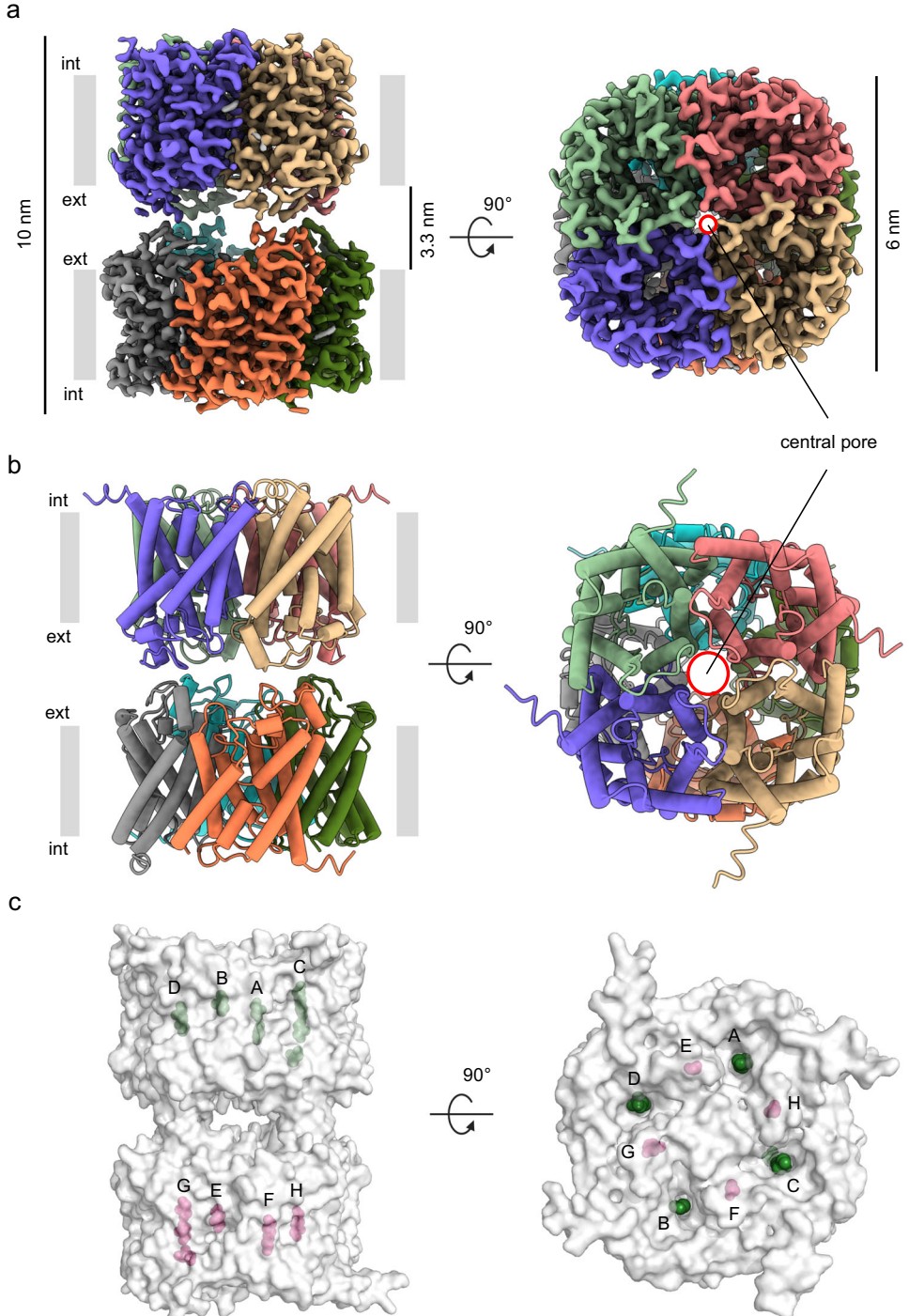

**Fig. 2 | Structural overview of the two adhering tetramers of AQP7 determined by cryo-EM. a** Side and top views of the 3.0 Å map of AQP7, generated by applying C1 symmetry. The map is colored by chain. **b** Side and top views of the model were generated using the 3.0 Å data. The model is shown as cylindrical helices and colored by chain. The central pore in the top view (from the cytoplasm) is marked by red circles in **a** and **b**. **c** Side and top views of the 3.0 Å model are shown as surface. Modeled glycerol molecules are represented by green spheres in chains A-D and magenta spheres in chains E-H, indicating the position of the four separate glycerol channels per tetramer.

of TM2 and TM5 from each of the four monomers, the central pore is wider in the cryo-EM structure compared to the X-ray structure (Fig. 5c). This results in a slightly shifted tetramerization, which potentially could be a result of AQP7 being vitrified for cryo-EM data collection, thus presumably in a less restrictive environment than upon crystallization (Fig. 5c). Analyses using the software HOLE of the central pore suggest that two quadruplets of leucine residues constrict the pore towards the extra- and intracellular sides at the position of

residue 68 (TM2) and residue 204 (TM5) (radii of 0.97 Å and 1.75 Å, respectively), creating two cavities with heights of ~18 Å (Fig. 6a, b). In contrast to the crystal structure, in both AQP7 single-particle cryo-EM maps (3.0 Å and 2.55 Å), clear, non-proteinous density features can be identified in the central pore cavities, in the vicinity of L72 and F76 (Fig. 6a and Supplementary Fig. 11). However, as four-fold symmetry has been applied to the 2.55 Å map, the density of the central pore cannot be analyzed using this map, and instead non-symmetry applied

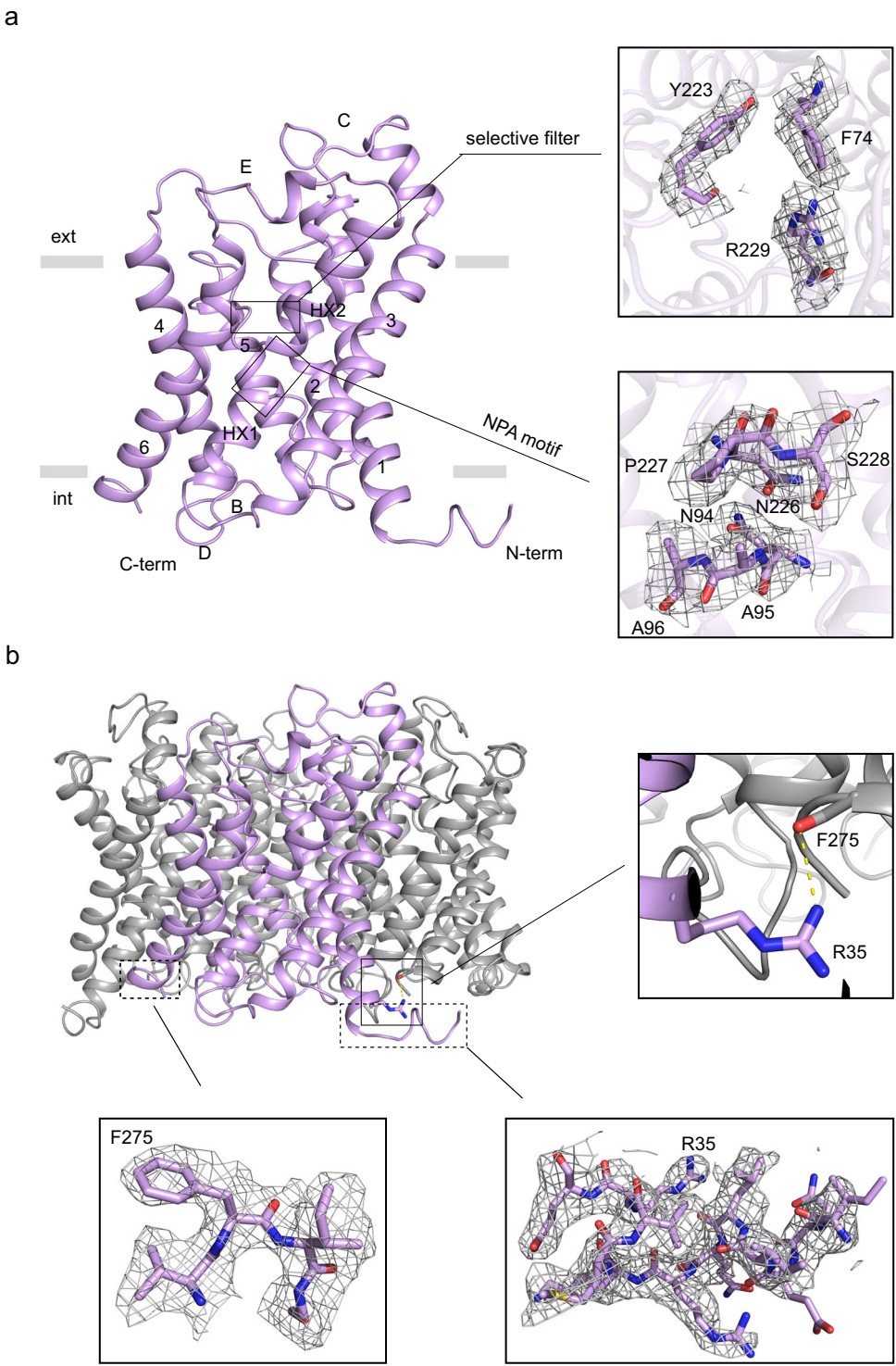

**Fig. 3 | The termini stabilize the tetrameric configuration of AQP7. a** Cartoon representation of the consensus monomer. Residues of the ar/R selective filter and NPA motifs are shown as sticks with corresponding cryo-EM density maps in gray. Transmembrane domains are represented by 1–6 while the two half helices by HX1 and HX2, and loops by A-E. **b** Interaction between neighboring monomers (A and C) in the 2.55 Å model. R35 in chain A and F275 in chain C are shown as sticks, and a H-bond is represented by a yellow dashed line in the right zoom-in view. The modeled N- and C-termini with corresponding cryo-EM density maps are shown at the bottom. The same contour level was adopted for the cryo-EM map in **a** and **b**.

3.0 Å map and model are used for assessing the central pore. To elucidate the origin of these densities, we investigated the surface charge using the APBS tool in Pymol, which suggests an overall positively charged environment, despite the hydrophobic character of the residues nearby (Fig. 6c). Next, the AQP7 cryo-EM sample was analyzed using GC/MS. The analyses showed the presence of glycerol and 18-carbon fatty acids such as oleic acid (Fig. 6d), which correlates well with the structural results as 18-carbon fatty acids fit well in the cryo-EM densities at the surface of transmembrane domains (Supplementary Fig. 12). In addition, the metabolites Gro3P and glycerol-2-phosphate (Gro2P) were detected in the sample (Fig. 6d), displaying similar molecular size as the cryo-EM densities in the central pore, and

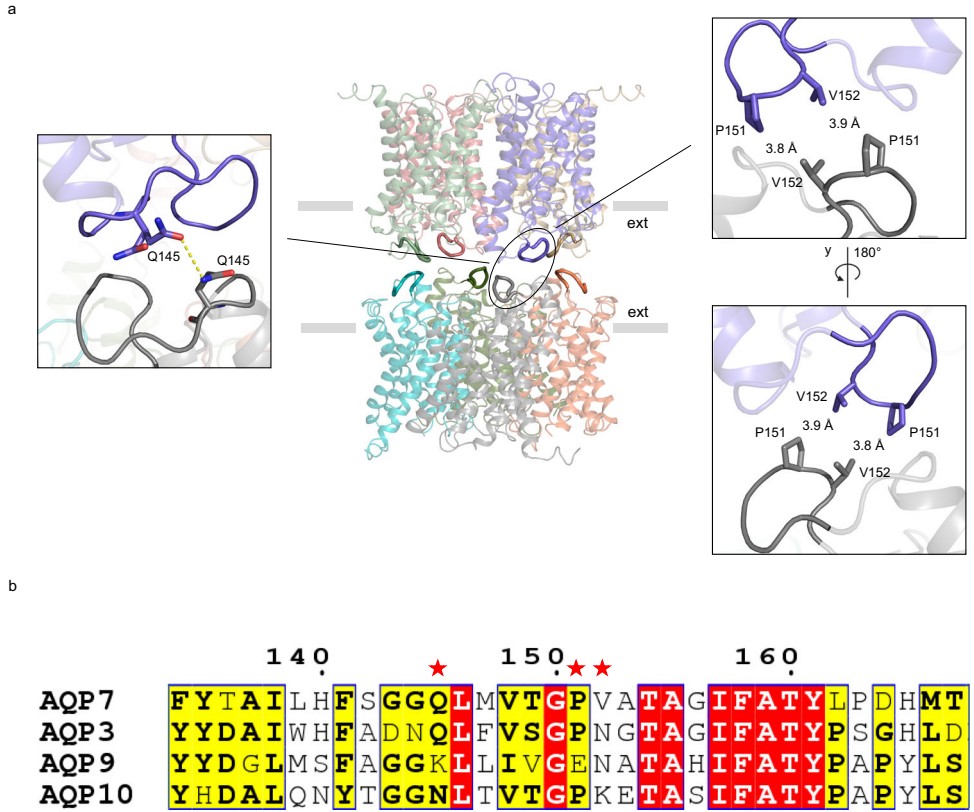

**Fig. 4 | AQP7 tetramers adhere via loop C. a** Cryo-EM structure shown as cartoon and the loop C of each monomer is highlighted as tube, contributing residues as sticks and hydrogen bonds shown as yellow dashed lines. **b** Sequence alignment by CLUSTALW (version 2.1) of loop C among the human aquaglyceroporins. Residues important for the octamer formation, Q145, P151 and V152, are marked by red stars, and highly and relatively conserved residues are shadowed in red and yellow, respectively.

amphipathic character congruent with the characteristics of the central pore.

## Discussion

To date, all available high-resolution structural information of AQPs have been obtained by crystallographic techniques applying either X-ray crystallography (AQP1, AQP2, AQP4, AQP5, AQP7 and AQP10)[15–20] or electron crystallography (AQP0, AQP1 and AQP4)[14,21,22]. A disadvantage with crystallographic techniques is that the prerequisite is to obtain a crystal of the protein of interest, which may affect the final structure and hence functional interpretation. To overcome this potential artifact and shed light on the function of the central pore, a single-particle cryo-EM structure of AQP7 was determined to 2.55 Å resolution.

The potential function of the central pore for aquaporins is unclear, but several suggestions have been discussed, like ion conduction for AQP1[23–26] and/or a possible pathway for small, gas molecules, such as $CO_2$ and $O_2$[27]. Additionally, in the human AQP5 X-ray structure a lipid occupies the central pore and thereby prevents any substrate transport, and a similar scenario was found for the bacterial AQPZ[19,28]. None of these molecules fit the densities seen in the pore of AQP7 cryo-EM structure. Instead, among the experimentally detected candidates identified using GC/MS, the small intermediate metabolite standing in the crossroad connecting glucose and lipid metabolism[29], Gro3P, was detected in a significant amount and well-fitted in the cryo-EM density of the central pore of AQP7 (Fig.7 and Supplementary Fig. 13), while the Gro2P in similar size as Gro3P is not possible to fit into the density (Supplementary Fig. 14). The potential ligand is trapped in-between two narrow segments of the central pore created by leucine residues. Similar leucine gates have previously been observed for the glutamate-gated chloride channels, where the chloride ion is trapped within the channel lined by leucine residues with no apparent coordination[30], like Gro3P observed here.

Intriguingly, AQP7 forms the dimer of tetramers in the cryo-EM structure, stabilized by interactions via loop C. Previously, it was reported that both AQP0 and AQP4 can form 2D crystal layers by extracellular loop mediated interactions between tetramers in opposing membranes relevant for intercellular cell adhesion/junctions[14,21]. AQP0 has been shown to have dual roles in the human body; a structural role as a cell–cell adhesion molecule in the lens of the eye as well as an additional function as water channel[31]. Similarly, AQP4 has been reported to have dual functions in the blood-brain barrier interface[31]. AQP0 tetramers interact with each other in the crystal packing via loop A–loop A packing (Supplementary Fig. 9a and c), and termini cleavage mediates rearrangement in loop A resulting in AQP0 junction formation[13,21]. Rather than being stacked as AQP0, one AQP4 tetramer interacts with four tetramers in the opposite layer by the extracellular loop C of each monomer (Supplementary Fig. 9b and d). In contrast to AQP0 and AQP4, AQP7 tetramers are arranged in a twisted stacked manner and pack with the protruding loop C through residues Q145, P151 and V152 from all eight monomers, and thus facilitate the octamer formation. Thereby, it is rational to speculate that AQP7 dimer of tetramers, in analog to AQP0 and AQP4, may serve as junction proteins and promote the rosette-like structures formation of β-cells around blood capillaries[32], in addition to its function to facilitate intercellular distribution of glycerol and possibly other metabolites such as Gro3P. Cell–cell as well as cell-capillary contacts mediated by intercellular junctions in β-cells play an important role for the insulin containing vesicles to be recruited and fused with the plasma membrane to release insulin[5,33]. Along these lines, AQP7 was

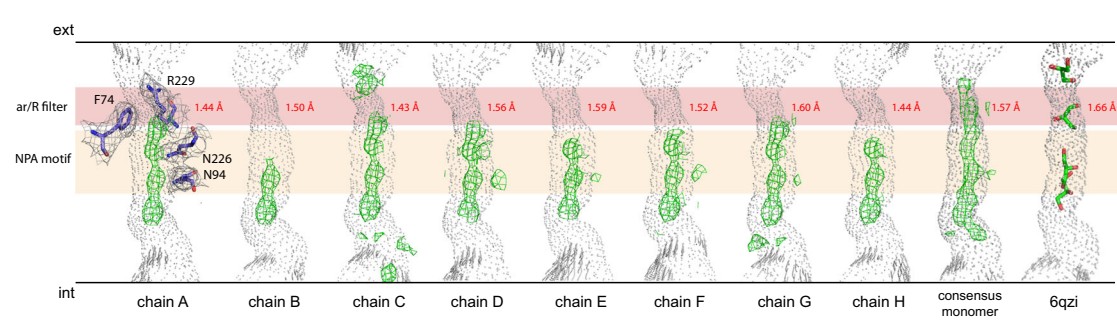

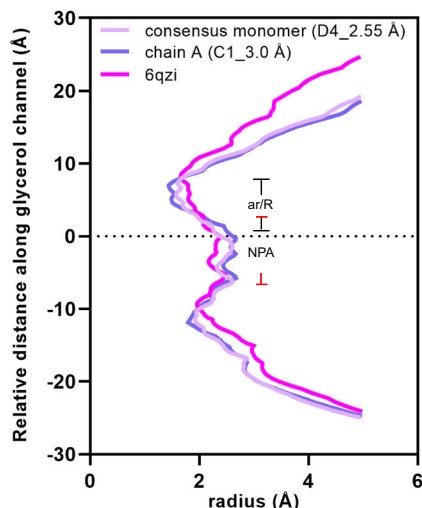

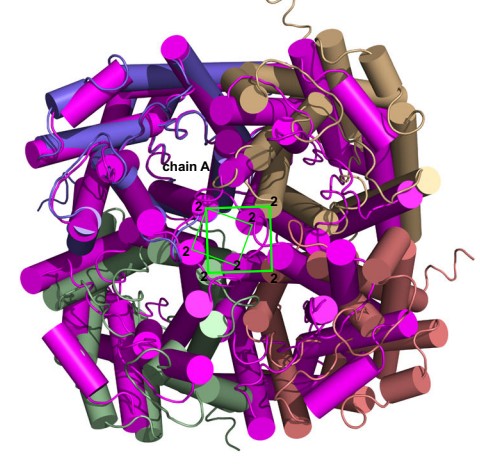

**Fig. 5 | Glycerol channels in AQP7. a** Glycerol channel profiles calculated using the software HOLE of all eight chains from the 3.0 Å (C1 symmetry) AQP7 cryo-EM model, the consensus monomer from the 2.55 Å (D4 symmetry) cryo-EM model, and the AQP7 crystal structure (PDB ID: 6qzi). Cryo-EM densities corresponding to glycerol molecules are shown as green mesh, while glycerol molecules from the crystal structure are shown as sticks in green. The regions of the ar/R selective filter and the NPA-motifs are marked in light red and light brown, respectively. The residues involved in ar/R and NPA regions in chain A are shown as sticks

accompanied by their cryo-EM densities in gray mesh. The same density contour level is applied for the residues and all glycerol molecules in the channels. **b** The radii along the glycerol channel (Å) were plotted for the consensus monomer, chain A of the 3.0 Å cryo-EM structure and the crystal structure (PDB ID: 6qzi). **c** Structural alignment between AQP7 3.0 Å cryo-EM structure colored by chain and AQP7 crystal structure in magenta shown as cylindrical helices. Chain A (purple) of cryo-EM structure is aligned with one monomer of crystal structure. Transmembrane helix 2 (TM2) in both structures are marked and connected by green lines.

recently reported to contribute to cell–cell interactions between islet cells, and impaired AQP7 expression levels were connected to pancreatic β-cell dysfunction[4]. Moreover, AQP7 is known to highly expressed in human capillaries[6]. Here we report that AQP7 is predominantly expressed in human β-cells and is forming dimer of tetramers, thereby supporting a role for AQP7 in cell–cell adhesion as well as promoting the structures formed by β-cells around blood capillaries. In addition, we report that the AQP7 expression is reduced in islets from T2D donors, suggesting that reduced AQP7 activity contributes to the islet dysfunction observed in T2D. We show that AQP7 expression is also negatively correlated with BMI, a condition that contributes to β-cell dysfunction. Thus, AQP7 expression levels, like other β-cell marker proteins, may be reduced in T2D, resulting in elevated BMI due to metabolic stress induced by increased fat mass. A critical role for the AQP7 gene locus in controlling islet cell function is further supported by the presence of genetic variants influencing blood glucose levels (HbA1c, T2D, fasted and random blood glucose), supporting a role for this genomic region in metabolic diseases, but more evidence is required to localize the causal variants. Taken together, as glycerol and Gro3P are essential molecules for β-cell function, and their availability controls insulin secretion[34,35], furthermore, cell–cell/capillary junctions in β-cells are suggested to be important for

the regulation of insulin secretion[5,33], the suggested dual activity of AQP7 (metabolite facilitator and junction protein) is likely to be crucial for a proper function and energy homeostasis of the human endocrine pancreas.

## Methods

### AQP7 expression analysis in human pancreatic islets

Gene expression from 188 donors was processed as described previously[36]. Human pancreas was obtained from the Human Tissue Laboratory, which is funded by the Excellence Of Diabetes Research in Sweden (EXODIAB) network (www.exodiab.se/home) in collaboration with The Nordic Network for Clinical Islet Transplantation Program (www.nordicislets.org). Informed consent was obtained from pancreatic donors, or their relatives and all procedures were approved by the Swedish Ethical Review Authority (Permit number 2011263).

Briefly, post-processing of expression data was performed by alignment to reference genome build 37, and gene counts were computed using feature counts. Counts were further normalized for sequencing depth and rank-based inverse normal transformation was applied. Linear models were applied to assess the association between gene expression and genotypes with age, sex, BMI, purity, and days in culture as covariates. GWAS look-ups of T2D and related traits (HbA1c,

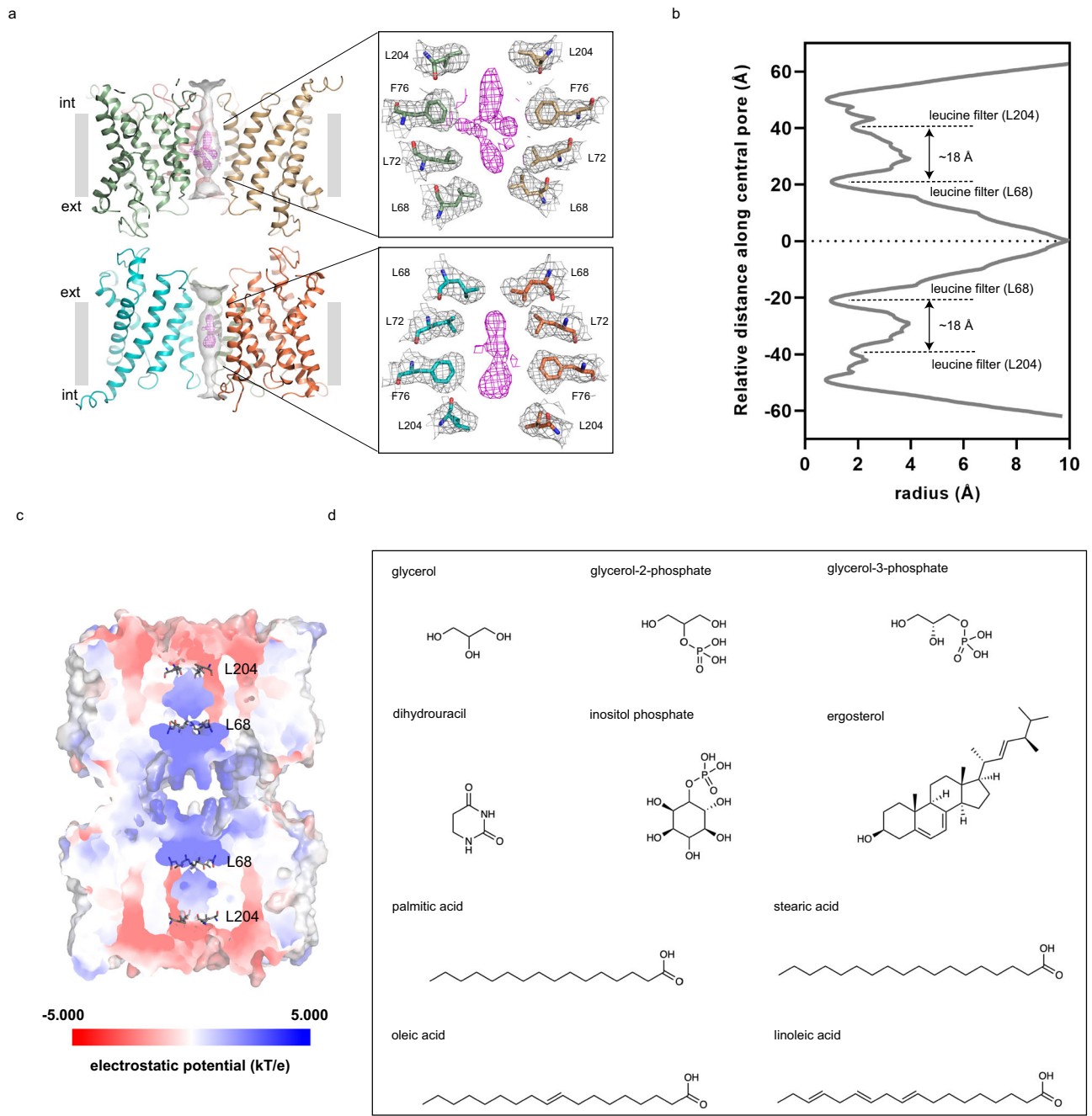

Fig. 6 | The central pore of AQP7 show well defined densities. a Software HOLE analysis of the central pore shown as gray surface, the AQP7 octamer as cartoon, non-proteinous cryo-EM density in the central pore in magenta shown in the same contour level as the pore lining residues. b The radius of the central pore along the pore axis. The central cavity of each tetramer is limited by two quadruplets of leucines (L68 and L204) separated by 18 Å. c APBS electrostatic analysis of the AQP7 octamer displayed as surface. The leucine filters are shown as sticks. d The molecules and corresponding structures identified in the AQP7 protein sample by GC/MS. The structures are drawn by ChemOffice (version 22).

fasted and random blood glucose levels) at the *AQP7* locus were performed in the AMPT2D portal (Type 2 Diabetes Knowledge Portal−Home (hugeamp.org)). Single cell sequencing data from Segerstolpe et al.[37], were re-analyzed to assess distribution of *AQP7* gene expression in islet cells.

### Immunohistochemistry
Immunohistochemical analysis was performed as described previously[38]. Briefly, human pancreas biopsies from normoglycemic donors were fixed in 4% paraformaldehyde and embedded in paraffin and 6-μm sections were mounted on glass slides. The primary antibodies with the following dilutions were used: insulin (1:1000. Dako, Cat#506442), AQP7 (1:200. Abcam, Cat#ab32826), glucagon (1:1000. Abcam, Cat#ab10988) and somatostatin (1:200. Abcam, Cat#ab30788). Secondary antibodies used were: Cy2-, Cy3-, and Cy5-conjugated α-guinea pig, α-mouse, α-rat, and α-rabbit (1:500. Jackson ImmunoResearch, Cat#s: 711-165-152, 712-176-153, 706-225-148, 715-225-150, 715-175-150). All antibodies have been validated by respective manufacturers at least by Western blot. The secondary antibodies have been routinely used in our lab and produce reliable results with multiple primary antibodies. All antibodies have also been reported in previous studies with different species (including human and mouse)

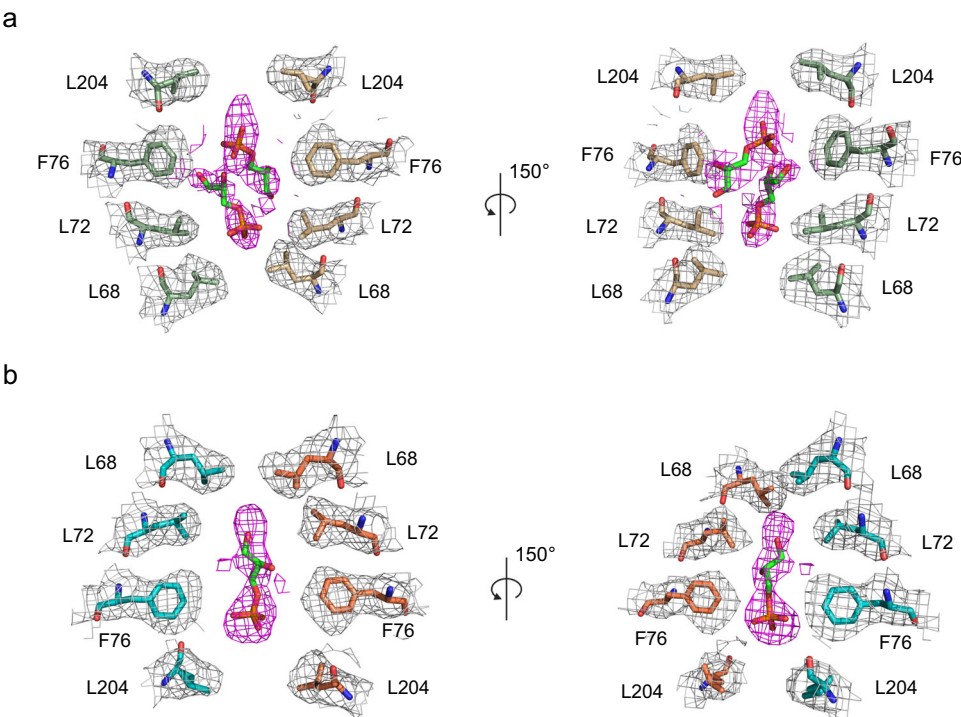

**Fig. 7 | Gro3P fitting into the density in the central pore.** The central pore formed by chain A-D (**a**) and E-H (**b**). The Gro3P molecules are shown as green sticks while the density in the central pore is shown as magenta mesh in the same contour level as lining residues density in gray. The molecules are shown in two different views.

and we have confirmed expression patterns and subcellular localization. Nuclear staining was performed using DAPI (1:6,000. Invitrogen).

## AQP3/7 preparation for cryo-EM

Full-length human aquaglyceroporin 3/7 (AQP3/7) genes fused with C-terminal poly-histidine tag was cloned into the double-deletion strain *P. pastoris GS115 aqy1Δ::HIS4 agp1Δ::NatMX* with no endogenous aquaporins and aquaglyceroporins present[39]. Yeast cells of AQP3 and AQP7 constructs were cultured and membranes were prepared and solubilized according to published protocols with minor modification[40]. Membranes were solubilized in the buffer (20 mM sodium phosphate pH 8.0, 300 mM NaCl, 10% glycerol, 1 mM DTT, EDTA-free protease inhibitor cocktail (Thermo Fisher Scientific) supplemented with 1% n-Dodecyl-β-D-Maltopyranoside (DDM, Anatrace) for AQP3 (room temperature, 1 h) or 1% n-Decyl β-D-maltopyranoside (DM, Anatrace) for AQP7 (4 °C, 1 h), respectively. After ultra-centrifugation, the supernatant for AQP3/AQP7 was diluted with solubilization buffer and incubated with Ni-NTA resin (Invitrogen) overnight at 4 °C. Protein purification was performed by slow washing of Ni-NTA with solubilization buffer supplemented with 0.25% glycoldiosgenin (GDN, Anatrace) and washing buffer containing 20 mM imidazole and 0.02% GDN, consecutively. Protein of interest was eluted by buffer containing 20 mM sodium phosphate pH 8.0, 500 mM NaCl, 20% glycerol, 300 mM imidazole and 0.02% GDN, followed by size-exclusion chromatography (Superose 6 Increase 10/300 GL, Cytiva) in the buffer of 20 mM sodium phosphate pH 7.5, 100 mM NaCl and 0.01% GDN. Corresponding peak fractions from size-exclusion chromatography were pooled and concentrated for cryo-EM grids preparation the next day.

## Cryo-EM data acquisition and processing

3 µl of 1 mg/mL AQP7 sample was applied onto glow-discharged holey carbon grids (60 s, 20 mA, Quantifoil Cu R1.2/1.3, 300 mesh), blotted for 3 s and plunge-frozen in liquid ethane using a Vitrobot MarK IV (FEI) at 4 °C and 100% humidity. Grids were transferred to a Titan Krios

electron microscope (FEI) operating at 300 kV equipped with spherical aberration (Cs) image corrector. Micrographs were recorded using a K3 Summit direct electron detector in super-resolution mode. Each stack of 40 frames was exposed for 2 s, and the total dose for each stack was ~50 e − /Å2. All 40 frames in each stack were aligned, summed, dose weighted and twofold binned to a pixel size of 0.8464 Å. Finally, ~14,000 micrographs were collected using the parameters above.

The collected dataset was divided into 4 data subsets (1–4) manually, including 3168, 5691, 2631 and 3225 movies, respectively. The image motion was corrected, and contrast transfer function (CTF) was estimated for all datasets in cryoSPARC[12]. After manual inspection to discard poor micrographs, 2566, 3865, 1827, 2995 micrographs were left in each subset for particle picking. Firstly, particles were extracted from 800 micrographs in dataset 1 to perform 2D classification, from which good class averages were selected as templates for particle picking from all 2,566 micrographs with a box size of 400 pixels in dataset 1. The picked particles were extracted and classified using 2D classification and good particle classes with different potential views of protein were selected as templates for particle picking in the other 3 subsets as well as the whole dataset, respectively. For each dataset, the ideal particles from 2D classification were divided into 5 classes on the 3D level using ab-initio reconstruction followed by heterogeneous refinement individually. The particles and volume presenting dimerized protein projection was picked and further subjected to non-uniform refinement imposing by D4 or C1 symmetry, respectively, yielding the best resolution maps from dataset 4, at 2.7 Å (D4 symmetry) and 3.2 Å (C1 symmetry). Particles corresponding to these two maps were further refined iteratively with global CTF refinement followed by local CTF refinement, respectively, before applying non-uniform refinement again, thereby improving the map resolution to 2.55 Å (D4 symmetry) and 3 Å (C1 symmetry), respectively, based on the gold-standard Fourier shell correlation (GSFSC) with a cut-off at 0.143. All maps were polished using the sharpening tool in cryoSPARC before model building.

In addition, AQP7 and AQP3 protein samples in GDN were also flash frozen on C-flat grids (C-flat Cu R1.2/1.3, 300 mesh) using the

same conditions as described above. Two grids were subjected to data collection, before the data was processed by cryoSPARC, respectively.

## Model building and refinement

The AQP7 crystal structure was fitted into the cryo-EM map in Chimera[41], after which the model was refined by real space refinement in Phenix[42] by restricting secondary structure and geometry. Then the model corrections, including clashes as well as rotamer and Ramachandran plot issues were performed manually using coot[43]. Potential ligands (glycerol, glycerol 3-phosphate and water) were added to the 3.0 Å model using coot, and, likewise, water molecules were added to 2.55 Å model using the Douse tool in Phenix. Glycerol 3-phophate was fitted into the density in the central pore of the 3.0 Å model to explore the possibility of this ligand occupying the pocket of the central pore, however for the final models submitted to the PDB (PDB IDs 8AMX and 8AMW for the 2.55 Å and 3.0 Å structures respectively) the ligand in the central pore was not included, as we cannot exclude other molecules are present in the pore. Finally, these models were subjected to real space refinement in Phenix and statistics calculation.

## HOLE analysis of pores

The HOLE software was downloaded from http://www.holeprogram.org/[44] and the dimensions of central pore and glycerol channels were calculated. HOLE analysis was performed with end radius of 10 Å on the model of a dimer of tetramers and 5 Å on monomers separately, after removing all hetatoms in the model. All HOLE profiles were visualized by Pymol, and the relative distance along the central pore or glycerol channel was plotted against the radius in GraphPad Prism.

## Gas chromatography/mass spectrometry analysis

Sample preparation was performed according to the protocol described previously[45]. In short, 0.1 ml of the samples were dried using a miVac concentrator (SP Scientific) for 3 h at 30 °C. Dried samples were methoximated with 20 μl of methoxyamine hydrochloride (Thermo Scientific) in pyridine by shaking at 1000 rpm for 30 min at room temperature (VWR, VX-2500). Then, 20 μl of N-methyl-N-(trimethylsilyl) trifluoroacetamide (MSTFA) + 1% trimethylsilyl chloride (Thermo Scientific) was added to each sample following shaking at 1000 rpm for 30 min at room temperature. Finally, derivatized samples were transferred into glass vials equipped with micro-inserts, capped, and immediately analyzed by gas chromatography/mass spectrometry (GC/MS).

An Agilent 7890 A (Agilent Technologies) gas chromatograph equipped with an HP-5MS (Agilent Technologies) (30 m length × 250 μm internal diameter; 0.25 μm film thickness) coupled to an Agilent 5975 C VL MSD mass spectrometer (Agilent Technologies) was used to analyze the samples. The system was controlled by MassHunter Acquisition software (version 10.0). A volume of 1 μl was injected at the temperature of 270 °C. The helium gas flow rate was 1 ml/min with a temperature gradient starting at 70 °C for 2 min, increasing 15 °C/min to 320 °C where it was kept constant for 2 min. Single ion monitoring was performed at m/z 357 Da (glycerol 3-phosphate 4TMS) while full scan was monitored between 50 and 550 Da. Electron ionization was conducted at 70 eV with a source temperature of 230 °C and a quadrupole temperature of 150 °C. Deconvolution was conducted in ChemStation (version 1.0) and mass spectra similarity and the retention index for glycerol 3-phosphate were confirmed by the NIST library (version 2) and authentic standard.

## Ethics

Human pancreas was obtained from the Human Tissue Laboratory, which is funded by the Excellence of Diabetes Research in Sweden (EXODIAB) network in collaboration with The Nordic Network for Clinical Islet Transplantation Program. Informed consent was obtained from pancreatic donors of their relatives and all procedures were approved by the Swedish Ethical Review Authority (permit number 2011263).

## Reporting summary

Further information on research design is available in the Nature Portfolio Reporting Summary linked to this article.

## Data availability

The data that support this study are available from the corresponding authors upon reasonable request. The cryo-EM maps have been deposited in the Electron Microscopy Data Bank (EMDB) under accession code EMD-15528 (AQP7 D4 symmetry) and EMD-15527 (AQP7 C1 symmetry). The coordinates have been deposited in the Protein Data Bank (PDB) under accession codes 8AMX (AQP7 D4 symmetry) and 8AMW (AQP7 C1 symmetry). Previously published PDB codes referred to in this paper are 6QZI (AQP7 X-ray structure), 2B6O (AQP0 X-ray structure) and 2D57 (AQP4 X-ray structure). Source data underlying Fig. 1b–d are provided as a Source Data file along this paper. Source data are provided with this paper.

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

## Acknowledgements

The cryo-EM data was collected at the Cryo-EM Swedish National Facility funded by the Knut and Alice Wallenberg, Family Erling Persson and Kempe Foundations, SciLifeLab, Stockholm University and Umeå Uni-versity. We would like to thank Lina Gefors at LBIC, Lund University for negative stain of the sample, and Julian Conrad and Marta Carroni in SciLifeLab in Stockholm University for cryo-EM grid preparation and data collection, and Sofia Essen for sample preparation and running the GC/MS. We would also like to thank Pål Stenmark at Stockholm University for critical discussions of the structural data. This study was supported by the Swedish Research Council (2020-02028, 2017-05816 to K.L.-P.), Diabetesfonden (DIA2020-500 to K.L.-P.), Swedish Cancer Society (20 0747 PjF) and Chinese Scholarship Council (CSC file NO: 201706170065 to P.H.).

## Author contributions

P.H., R.V., I.A. and K.L.-P. conceived the study. P.H., R.V., R.B.P., H.A., S.W.d.M., X.F., P.S., N.Y., I.A and K.L.-P. designed the experiments. R.P. and I.A. performed cell biology experiments. P.H. and S.d.M. prepared protein samples and cryo-EM grids for structural study. P.H. and X.F. performed cryo-EM data collection and process. P.H., R.V and P.L. built and refined structural models. H.A. and P.S. performed mass spectro-metry experiments. All authors analyzed and interpreted the results. P.H., R.P. H.A. and I.A. prepared figures in which P.H made all figures relevant with protein structures. P.H., R.V., P.G., I.A. and K.L.-P. wrote the manuscript.

## Funding

## Competing interests

The authors declare no competing interests.
