## [Peer Review File · Nature Communications]

Cryo-EM structure supports a role of AQP7 as a junction proteinReviewers' Comments:

Reviewer #1:

Remarks to the Author:

The manuscript entitled "Cryo-EM structure supports a role of AQP7 as a junction protein" by Huang et al. describes structural and functional assessment of AQP7. Of major interest is the observation that AQP7 oligomerizes in a manner not yet observed for this aquaporin. This oligomer potentially explains cell adhesion function observed by other researchers. Although a nice contribution to the field, several parts of the work could use expanding on in greater detail. Of highest interest and priority are experimental validation of the interactions observed in cryo-EM that are purported to enable cell adhesion function. As only a fraction of the total particles, strong experimental evidence is thus lacking. Specific items to address are described below.

Figures

- Fig 4b could benefit from better coloring and demarcations highlighting the homologous from divergent regions.
- Fig 5a: the densities for glycerol look remarkably consistent for chains E-H while for chains A-D in the other tetramer they vary. Is there an explanation for this?
- Fig 5b: it was not immediately evident that this was an analysis of the central pore. This should be added to the figure.
- Fig 5c: the crystal structure has 1 mol in the au and thus no major structural deviations between tetramer subunits exist. In the cryo-EM structure, are individual monomers of the tetramer structural distinct? A structural alignment or RMSD calculation table may help to reveal this to readers. Perhaps in the Extended Data. To clarify, what I'm requesting is a monomer to monomer comparison of the cryo-EM tetramer and not the tetramer to tetramer comparison shown here. This is not to say the latter is not helpful, it should be kept as well.
- For Fig 5, it would be important to add and compare the 2.55 Å structure in these analyses too, in order to provide a sense of the general error or normal dynamic fluctuations in similar structures. While I understand the rationale of focusing on the 3 Å structure, the 2.55 Å structure can be viewed as a validation tool.
- Fig 6e: what is known about how molecules may move through the pore, especially their orientations? Is it expected for them to traverse the pore in random orientations with their phosphates in opposite directions? The manner in which G3P traverses the pore depicted does not seem coordinated or controlled. Please provide a rationale or reference for this, if possible.
- Fig 6e-f: is there a reason these figures are mirror images of each other? Makes it harder to envision with one being upside-down.

Results

- Line 186: "the central pore is wider " at some points yes but at others narrower. This sentence requires clarification.
- Line 188: "being in solution upon cryo-EM data collection and thus in a more natural environment ". While the sample was in solution, it was then frozen on grids into ice. This is not a more natural environment. Less restrictive than crystallization perhaps but not natural. Please re-word.
- Are G3P present in both resolution structures, are there differences or similarities? Like suggestion for Fig 5, a comparison may be warranted.
- Extended Data Fig 2b: Most if not all the red squared particles, which appear to be dimer tetramers, are side orientations alone. Can it be verified that any top down particles actually contain 2 tetramers? Some particles not squared appear as double stacked dimer tetramers, thus this orientation may be induced by GDN interactions with the grid and have not physiological relevance. It would be expected that if these interactions occurred in solution with any relative affinity that SEC traces would show more octamer. This does not appear to be the case. This should be addressed, in Experimental Suggestions (below)

Discussion

- Discussion paragraph 3: the words "weakly" (Line 248) when referring to AQP4 and "stable" (Line 252) for AQP7 are unjustified without binding data to verify. The structures suggest the AQP4

interactions are likely not weak while AQP7 octamers are only a fraction of the total particles and thus not stable. Please re-word or omit such relative terms.

- Discussion paragraph 3 and paragraph 4: As it's the Discussion it is fine to speculate about cell adhesion function of AQP7 but it should be noted that this paper does not provide functional evidence of cell adhesion. Cell adhesion proteins typically have very intricate intermolecular arrangements with large intermolecular interfaces. The AQP7 structure shown here would thus appear to lack these traits, especially compared to AQP0 and 4 (Extended Data Fig 8). In the case of AQP4, the blood-brain barrier is mentioned, which is a true cell adhesion structure in endothelium. Thus, AQP4 could just as likely take advantage of close cellular contacts facilitated by other cell adhesion molecules and have no adhesion function unto itself. A sentence stating more research into cell adhesion function should help to validate this hypothesis, is suggested.

Methods

- Model Building: I don't understand the rationale for not depositing the structures with the ligand G3P. If the density is good enough to model it for figures and this manuscript it should be good enough for deposition. If it is not, then it should not be discussed in this manuscript. Please justify the rationale for using the Results section to show this ligand binding capacity but not depositing it with the structures. If this is truly speculative perhaps it should be moved to the Discussion and Extended Data. It's inclusion does not bolster to cell adhesion focus of the paper.

Tetramer/Tetramer Interaction

- For the dimer of tetramer, it appears as though these interactions have not been observed in crystal structures of AQP7. It would seem that if such an interaction were favorable in cryo-EM than via crystallization it would have been visualized as well. Please comment as to whether this is true and/or provide a rationale for why this oligomerization may only be capable of forming in cryo-EM. The explanation given in Line 187 is insufficient. See bullet point 1 in Discussion.

- Extended Data Fig 2a: Based on elution volume from SEC the protein appears as a mono disperse homotetramer? If so, would this homo-tetramer interaction not appear to be concentration dependent perhaps or a function of the grid chemistry?

Experimental Suggestions

- Any thought of 1) making mutations to loop C residues involved in trans interactions and test binding via a biosensor based assay or negative stain to assess results on octamer formation?
- The major question that exists is whether this octamer is a result of physiological interactions trapped on a cryo-EM grid or if they are detergent/grid induced. Mutation of the interface residues may answer this question. SEC may not change but if mutated interface residues lack these sandwich shaped structures it could be reasonably concluded that detergent/grid interactions are not inducing these protein directed interactions. Such experiments could be done using negative staining EM or quantified using biophysical techniques like SPR or BLI. Such validation would strongly support the discussion and conclusions of this manuscript, and could be completed in a relatively short amount of time with minimal effort.

Reviewer #2:

Remarks to the Author:

Nature Communications: NCOMMS-22-14541-T

Cryo-EM structure supports a role of AQP7 as a junction protein

Corresponding author: Karin Lindkvist-Petersson

The aquaporins and aquaglyceroporins are membrane-spanning proteins that exist as homo-tetramers in the lipid bilayer of native plasma membranes where they provide a pathway for rapid and selective movement of water or water + glycerol into and out of cells. The aquaporins and aquaglyceroporins have been studied in diverse tissues and are believed to be expressed by all living organisms. Previous studies of AQP7 in cultured cells and rodents provide insight into the physiology and pathophysiology, but study of human AQP7 has lagged.

In this interesting manuscript, the investigators pursued human AQP7 in pancreatic islet β -cells by cryo-EM and by GC/MS. Taken together the studies demonstrated molecular and structural determinants related to β -cell homeostasis. The technology is clearly described, and the text is clearly written and concise. Important features of the study include:

1. Identification of tetramer-tetramer association through determinants in loop C (octomer-junction) with a 45° twist.
2. Implication of glycerol-3-phosphate filling the heretofore mysterious central pore.
3. Correlating AQP7 expression with type 2 diabetes will likely attract significant interest from clinicians as well as biochemists and structural biologists.

Critical points:

1. On my computer screen, the two micrographs in Fig 1a do not show clear localizations of glucagon and somatostatin as stated in the text.
2. The identities of the points (black, gray, and cyan) in Fig 1d are not specified.
3. Additional discussion and speculation of how AQP7 junctions contribute to islet β -cell homeostasis could be helpful.

Peter Agre
Johns Hopkins Univ
Bloomberg School of Public Health

Point-by-Point response to the comments of the reviewers.

Reviewers comments in black and our response in blue

REVIEWER COMMENTS

Reviewer #1 (Remarks to the Author):

The manuscript entitled "Cryo-EM structure supports a role of AQP7 as a junction protein" by Huang et al. describes structural and functional assessment of AQP7. Of major interest is the observation that AQP7 oligomerizes in a manner not yet observed for this aquaporin. This oligomer potentially explains cell adhesion function observed by other researchers. Although a nice contribution to the field, several parts of the work could use expanding on in greater detail. Of highest interest and priority are experimental validation of the interactions observed in cryo-EM that are purported to enable cell adhesion function. As only a fraction of the total particles, strong experimental evidence is thus lacking.

We thank reviewer #1 for the nice summary of our work.

Specific items to address are described below.

Figures

- Fig 4b could benefit from better coloring and demarcations highlighting the homologous from divergent regions.

The coloring has been improved with highlighting the homologues and divergent regions.

- Fig 5a: the densities for glycerol look remarkably consistent for chains E-H while for chains A-D in the other tetramer they vary. Is there an explanation for this?

The reason for the consistency between the glycerol densities is most likely due to stronger interactions with the protein at certain parts of the channel, resulting in that the glycerol molecules will occupy those positions more often. This is correlating well with the crystal structure as we can see consistent density for glycerol at the same positions. That chains E-H are more similar than A-D, we believe is a coincidence.

- Fig 5b: it was not immediately evident that this was an analysis of the central pore. This should be added to the figure.

Fig 5b is not analyzing the central pore, it is showing the radii along the glycerol channels. To clarify this further, we have now edited the entire manuscript so that when we refer to the glycerol pores, we call them "glycerol channels" and when we refer to the central pore we call it "central pore". Labels have been added in the figure. Sorry for this unclarity.

- Fig 5c: the crystal structure has 1 mol in the au and thus no major structural deviations between tetramer subunits exist. In the cryo-EM structure, are individual monomers of the tetramer structural distinct? A structural alignment or RMSD calculation table may help to reveal this to readers. Perhaps in the Extended Data. To clarify, what I'm requesting is a monomer to monomer comparison of the cryo-EM tetramer and not the tetramer to tetramer comparison shown here. This is not to say the latter is not helpful, it should be kept as well. Thank you for this comment. The monomers in the cryo-EM structure are very similar. To show this the RMSD between the eight monomers are now available in the Extended Data Table 2. In addition, a comparison of the radii along the eight glycerol channels are in Extended Data Figure 10.

- For Fig 5, it would be important to add and compare the 2.55 Å structure in these analyses too, in order to provide a sense of the general error or normal dynamic fluctuations in similar structures. While I understand the rationale of focusing on the 3 Å structure, the 2.55 Å structure can be viewed as a validation tool.

Thank you for this suggestion, the 2.55 Å data (called consensus monomer) has been added to both Fig 5a and 5b.

- Fig 6e: what is known about how molecules may move through the pore, especially their orientations? Is it expected for them to traverse the pore in random orientations with their phosphates in opposite directions? The manner in which G3P traverses the pore depicted does not seem coordinated or controlled. Please provide a rationale or reference for this, if possible.

We agree with the reviewer that from the structure we cannot detect a specific coordination of the Gro3P. If we search for similar examples, we can refer to GluCl channel structure (pdb id 4TNW), where a chloride ion is trapped within the channel lined by leucine residues with no apparent coordination. However, it is likely that Gro3P makes water mediated hydrogen bonds, but at this resolution water molecules are not resolved. The oxygen atoms of the phosphate group of the Gro3P are located within the 6.6-6.9Å distance from the carbonyl oxygens of Leu204 (measured in the EFGH tetramer), so the presence of the water molecules in between would make an interaction possible.

This paragraph has been added in the discussion:

“Similar leucine gates have previously been observed for the glutamate-gated chloride channels, where the chloride ion is trapped within the channel lined by leucine residues with no apparent coordination (31), like Gro3P observed here.”

- Fig 6e-f: is there a reason these figures are mirror images of each other? Makes it harder to envision with one being upside-down.

Thank you for this comment. Figure 6e (old numbering) is showing the Gro3P density for the one tetramer (the upper), while 6f (old numbering) shows the density for the lower tetramer. We have merged old 6e/f with 6a to one figure (now called 6a), to make this figure clearer.

Results

- Line 186: “the central pore is wider ” at some points yes but at others narrower. This sentence requires clarification.

We refer to figure 5c in this case, and the central pore is wider overall compared to the X-ray structure. Just to clarify, Fig5b is showing the width of the monomer, not the central pore.

- Line 188: “being in solution upon cryo-EM data collection and thus in a more natural environment ”. While the sample was in solution, it was then frozen on grids into ice. This is not a more natural environment. Less restrictive than crystallization perhaps but not natural. Please re-word.

Thank for this comment the sentence has been rewritten and now reads:

“This results in a slightly shifted tetramerization, which potentially could be a result of AQP7 being vitrified for cryo-EM data collection, thus presumably in a less restrictive environment than upon crystallization (Fig. 5c).”

- Are G3P present in both resolution structures, are there differences or similarities? Like suggestion for Fig 5, a comparison may be warranted.

Figs 6e and 6f (old numberings) were showing the differences between the two tetramers, they have now been merged into one figure (Fig 6a) to clarify this for the reader.

- Extended Data Fig 2b: Most if not all the red squared particles, which appear to be dimer tetramers, are side orientations alone. Can it be verified that any top down particles actually contain 2 tetramers? Some particles not squared appear as double stacked dimer tetramers, thus this orientation may be induced by GDN interactions with the grid and have not physiological relevance. It would be expected that if these interactions occurred in solution with any relative affinity that SEC traces would show more octamer. This does not appear to be the case. This should be addressed, in Experimental Suggestions (below)

Thanks for your comments. Yes, in addition to side view dimers, we observe tilted views of dimers of tetramers in the 2D classification (called tilted views, Extended data Fig. 2c).

Furthermore, we obtained a 3D reconstruction from data process, suggesting that top-down view particles as well as side and tilted ones are present on the grid as dimers. To confirm the dimerization experimentally, we re-run SEC at different concentrations showing elution profiles with shorter elution times in addition to the longer elution times correlating to the tetramer size, suggestion that the dimer of tetramers exists in solution (Extended Data Figure 2b).

Discussion

- Discussion paragraph 3: the words “weakly” (Line 248) when referring to AQP4 and “stable” (Line 252) for AQP7 are unjustified without binding data to verify. The structures suggest the AQP4 interactions are likely not weak while AQP7 octamers are only a fraction of the total particles and thus not stable. Please re-word or omit such relative terms.

The words “weakly” and “stable” have been removed.

- Discussion paragraph 3 and paragraph 4: As it’s the Discussion it is fine to speculate about cell adhesion function of AQP7 but it should be noted that this paper does not provide functional evidence of cell adhesion. Cell adhesion proteins typically have very intricate intermolecular arrangements with large intermolecular interfaces. The AQP7 structure shown here would thus appear to lack these traits, especially compared to AQP0 and 4 (Extended Data Fig 8). In the case of AQP4, the blood-brain barrier is mentioned, which is a true cell adhesion structure in endothelium. Thus, AQP4 could just as likely take advantage of close cellular contacts facilitated by other cell adhesion molecules and have no adhesion function unto itself. A sentence stating more research into cell adhesion function should help to validate this hypothesis, is suggested.

Thank you for this comment. We agree and we have extended the paragraph concerning cell adhesion in the islet of Langerhans in the discussion.

Methods

- Model Building: I don’t understand the rationale for not depositing the structures with the ligand G3P. If the density is good enough to model it for figures and this manuscript it should be good enough for deposition. If it is not, then it should not be discussed in this manuscript. Please justify the rationale for using the Results section to show this ligand binding capacity but not depositing it with the structures. If this is truly speculative perhaps it should be moved to the Discussion and Extended Data. It’s inclusion does not bolster to cell adhesion focus of the paper.

Thank you for this comment, while the focus of the paper is indeed cell adhesion, we thought that it is important to address the cryo-EM information on the central pore, as its role is still not concluded, and it is an interesting question within the aquaporin-field. But we agree with

the reviewer that it is odd to have the data in the result section and not include it in the deposition, thus we have followed the reviewer's suggestion and moved the data concerning the fitting of the Gro3P into the cryo-EM density to the discussion and Extended data.

Tetramer/Tetramer Interaction

- For the dimer of tetramer, it appears as though these interactions have not been observed in crystal structures of AQP7. It would seem that if such an interaction were favorable in cryo-EM than via crystallization it would have been visualized as well. Please comment as to whether this is true and/or provide a rationale for why this oligomerization may only be capable of forming in cryo-EM. The explanation given in Line 187 is insufficient. See bullet point 1 in Discussion.

To explain why we do not see the dimer of tetramers in the crystals will of course be speculative, but one explanation for the lack of crystal packing type I (stacked membranes, instead of type II, which is formed) is that type I packing also requires interaction between the intracellular sides of AQP7 to be able to stack the molecules. This is probably physiologically very unlikely to happen and thus crystals with type I packing will not form. We note purification and crystallization conditions can also alter the multimeric state of membrane proteins, and thus it is possible that the conditions employed for crystallization disfavored the dimer of tetramers (PMID: 35857505).

- Extended Data Fig 2a: Based on elution volume from SEC the protein appears as a mono disperse homotetramer? If so, would this homo-tetramer interaction not appear to be concentration dependent perhaps or a function of the grid chemistry?

We have added more SEC runs at different concentrations where we can detect a varying degree of dimerizations. Concerning the grid-chemistry, we have tested another grid, and we could still detect dimers. As a control, we also tested AQP3, where no dimers could be seen (Extended Data Figure 4).

Experimental Suggestions

- Any thought of 1) making mutations to loop C residues involved in trans interactions and test binding via a biosensor based assay or negative stain to assess results on octamer formation?

We agree that it would be very interesting to analyze the physiological role of this dimerization by making mutations and possibly see differences in cell adhesion properties, however this is out of the scope for this manuscript. Still, instead of making mutations in AQP7 we analyzed the propensity for AQP3 (which could be seen as V152N substitution in loop C) to form dimers, and as can be seen in Extended Data Figure 4, no dimerization can be formed upon the V152N "substitution".

- The major question that exists is whether this octamer is a result of physiological interactions trapped on a cryo-EM grid or if they are detergent/grid induced. Mutation of the interface residues may answer this question. SEC may not change but if mutated interface residues lack these sandwich shaped structures it could be reasonably concluded that detergent/grid interactions are not inducing these protein directed interactions. Such experiments could be done using negative staining EM or quantified using biophysical techniques like SPR or BLI. Such validation would strongly support the discussion and conclusions of this manuscript, and could be completed in a relatively short amount of time with minimal effort.

To rule out that the dimerization is detergent/grid induced, we have analyzed AQP3 by cryo-EM. AQP3 is similar to AQP7 but has an asparagine instead of valine in loop C (Fig. 4b).

Applying the exact same conditions as for AQP7 (same detergent, grid type, blotting and glow-discharging conditions), it is clear from the 2D classes that AQP3 cannot form dimers (Extended Data Figure 4b). This suggests that the dimerization is not caused by the grid chemistry but represents an inherent feature of AQP7.

Reviewer #2 (Remarks to the Author):

Nature Communications: NCOMMS-22-14541-T
Cryo-EM structure supports a role of AQP7 as a junction protein
Corresponding author: Karin Lindkvist-Petersson

The aquaporins and aquaglyceroporins are membrane-spanning proteins that exist as homo-tetramers in the lipid bilayer of native plasma membranes where they provide a pathway for rapid and selective movement of water or water + glycerol into and out of cells. The aquaporins and aquaglyceroporins have been studied in diverse tissues and are believed to be expressed by all living organisms. Previous studies of AQP7 in cultured cells and rodents provide insight into the physiology and pathophysiology, but study of human AQP7 has lagged.

In this interesting manuscript, the investigators pursued human AQP7 in pancreatic islet β -cells by cryo-EM and by GC/MS. Taken together the studies demonstrated molecular and structural determinants related to β -cell homeostasis. The technology is clearly described, and the text is clearly written and concise. Important features of the study include:

1. Identification of tetramer-tetramer association through determinants in loop C (octomer-junction) with a 45° twist.
2. Implication of glycerol-3-phosphate filling the heretofore mysterious central pore.
3. Correlating AQP7 expression with type 2 diabetes will likely attract significant interest from clinicians as well as biochemists and structural biologists.

We thank reviewer #2 for the nice summary of our work.

Critical points:

1. On my computer screen, the two micrographs in Fig 1a do not show clear localizations of glucagon and somatostatin as stated in the text.

Thank you for this comment, we have clarified this with an additional inset showing glucagon and somatostatin.

2. The identities of the points (black, gray, and cyan) in Fig 1d are not specified.

Sorry for this mistake, we have added this to the legend.

3. Additional discussion and speculation of how AQP7 junctions contribute to islet β -cell homeostasis could be helpful.

The last paragraph in the discussion has been extended.

Peter Agre
Johns Hopkins Univ
Bloomberg School of Public Health

Reviewers' Comments:

Reviewer #1:

Remarks to the Author:

I thank the authors for carefully and fully addressing my suggestions. I believe the new manuscript and added experiments bolsters the major points and findings of the paper. Thus, I have no added comments. Congratulations on your work.

Reviewer #3:

Remarks to the Author:

I was asked to review the eQTL aspects of this work and am therefore only focusing on that part of the manuscript. Unfortunately, there are several problems from a statistical methods perspective. When assessing whether or not two statistical genetic signals overlap with each other - one from eQTL and the other from GWAS - the standard in the field is to perform statistical colocalization using one of a number of methods. For example, eCaviar and coloc are widely used methods. Neither are used here, and further, no statistical colocalization analysis is attempted. Therefore, there's no formal evidence that the AQP7 eQTL signal at rs10114322 is colocalized with a T2D GWAS signal or any of the other GWAS traits mentioned.

Point-by-Point response to the comments of the reviewers.

Reviewer comments in black and our response in blue

REVIEWER COMMENTS

I was asked to review the eQTL aspects of this work and am therefore only focusing on that part of the manuscript. Unfortunately, there are several problems from a statistical methods perspective.

Comment:

When assessing whether or not two statistical genetic signals overlap with each other - one from eQTL and the other from GWAS - the standard in the field is to perform statistical colocalization using one of a number of methods. For example, eCaviar and coloc are widely used methods. Neither are used here, and further, no statistical colocalization analysis is attempted. Therefore, there's no formal evidence that the AQP7 eQTL signal at rs10114322 is colocalized with a T2D GWAS signal or any of the other GWAS traits mentioned.

Thanks for your comments and suggestions. We have performed colocalization analysis and evaluated the overlap of a T2D GWAS signal, as requested. Unfortunately, this type of analysis requires a high statistical power and analysis of a large dataset. While we have analyzed RNA sequencing data from a substantial number of donors (n=188), this was not enough to obtain a statistically significant colocalization signal for AQP7 expression and T2D at the SNP in question (rs115571549). Thus, as the eQTL is a minor part of the manuscript and the presence of the results do not affect the overall conclusions of the manuscript, we have decided to remove the eQTL data in the manuscript.

Instead, we show more details for the gene variants at the *AQP7* gene locus that are associated with T2D and various parameters that measure blood glucose control. The paragraph in the result section now reads:

“To evaluate if genetic variants in *AQP7* gene locus were associated with T2D and metabolic traits associated with blood glucose control (HbA1c, fasting and random blood glucose measurements and HOMA-B), genome wide association data look ups were made. The rs2247654 variant in the *AQP7* gene showed strong signals of association with random glucose ($p=1.52*10^{-12}$) and fasting glucose adjusted for BMI ($p=5.08*10^{-12}$), and nominal association with HbA1c ($p=0.022$)⁸⁻¹⁰. Other variants in the *AQP7* gene showed suggestive signals for T2D (rs894105429: $p=3.98*10^{-4}$), HbA1c (rs139067302: $p=1.07*10^{-3}$) and HOMA-B (rs753118439: $p=8.37*10^{-3}$)¹¹ (AMPT2D portal: hugeamp.org) (Extended Data Fig. 1a). The SNPs rs83921 (~0.4 Mb distance of *AQP7*) and rs855532 (~0.25 Mb distance from *AQP7*) were significantly associated with T2D ($p=2.39*10^{-9}$)¹² and HbA1c ($p=7.42*10^{-8}$), respectively (<https://www.kp4cd.org/node/120>; AMPT2D portal: hugeamp.org) (Extended Data Fig. 1b).”

Thus, the robust GWAS signals for T2D and multiple related traits suggest that this genomic region is critical for metabolic control. Moreover, our data showing that AQP7 expression is present in human endocrine cells and that it is reduced in T2D islets further support that AQP7 is critical for maintaining physiological islet function.

Reviewers' Comments:

Reviewer #3:

Remarks to the Author:

It is good to have removed the GWAS-eQTL coloc results from the manuscript, as there was no support. I think these revisions are good, and I'd suggest further toning down levels of association. In GWAS, the genome-wide threshold for detecting a signal is 5×10^{-8} . If a variant does not pass this threshold, it's in the noise. So, some of the variants that are mentioned in the revised text that do not pass this threshold are null from a GWAS perspective. It's good to do this look ups in web portals, but please use caution when interpreting the results. Overall nice work, and thanks for removing the eQTL-GWAS coloc part.

Point-by-Point response to the comments of the reviewers.

Reviewer comments in black and our response in blue

Reviewer #3 (Remarks to the Author):

It is good to have removed the GWAS-eQTL coloc results from the manuscript, as there was no support. I think these revisions are good, and I'd suggest further toning down levels of association. In GWAS, the genome-wide threshold for detecting a signal is 5×10^{-8} . If a variant does not pass this threshold, it's in the noise. So, some of the variants that are mentioned in the revised text that do not pass this threshold are null from a GWAS perspective. It's good to do this look ups in web portals, but please use caution when interpreting the results. Overall nice work, and thanks for removing the eQTL-GWAS coloc part.

Thank you for these comments, we have removed all variants (figures and text) that do not pass the threshold suggested by the reviewer.